

# NORMS AND VALUES IN SOCIO-HYDROLOGICAL MODELS
Mahendran Roobavannan[1], Tim H. M. van Emmerik [2], Yasmina Elshafei[3], Jaya Kandasamy[1],
Matthew Sanderson[4], Saravanamuthu Vigneswaran[1], Saket Pande[2], and Murugesu Sivapalan[5, 6]
[1] School of Civil and Environmental Engineering, University of Technology Sydney, Sydney, NSW,
Australia
[2] Department of Water Management, Delft University of Technology, Delft, Netherlands
[3] School of Earth & Environment, The University of Western Australia, Crawley WA 6009,
Australia
[4] Department of Sociology, Kansas State University, Manhattan, KS 66506, USA
[5] Department of Civil and Environmental Engineering, University of Illinois at Urbana-Champaign,
Urbana IL 61801, USA
[6] Department of Geography and Geographic Information Science, University of Illinois at Urbana-
Champaign, Champaign IL 61820, USA
**ABSTRACT**
Sustainable water resources management relies on understanding how societies and water systems
co-evolve. Many place-based socio-hydrology (SH) studies use proxies, such as environmental
degradation, to capture key elements of the social component of system dynamics. Parameters of
assumed relationships between environmental degradation and the human response to it are usually
obtained through calibration. Since these relationships  are not yet underpinned by social science
theories, confidence in the predictive power of such place-based socio-hydrologic models remains
low. The generalisability of SH models therefore requires major advances in incorporating more
realistic relationships, underpinned by appropriate hydrological and social science data, and theories.
The latter is a critical input, since human  culture – especially values and norms arising from it -
influences behaviour and the consequences of behaviours. This paper reviews a key social science
theory that links cultural factors to environmental decision-making, assesses how to better
incorporate social science insights to enhance SH models, and raises important questions to be
addressed in moving forward. This is done in the context of recent progress in socio-hydrological



studies and the gaps that remain to be filled. The paper concludes with a discussion of challenges
and opportunities in terms of generalisation of SH models and the use of available data to allow
future prediction and model transfer to ungauged basins.

**KEYWORDS:** Socio-hydrology; culture; values and norms; modeling; content analysis.

## 1. INTRODUCTION
The concept of sustainable development has received much attention among researchers, policy
makers and stakeholders. Water is at the core of many of the sustainability challenges that human
societies face (Bai et al., 2016; Falkenmark and Rockström, 2004; Rijsberman, 2006). Sustainable
water resource management is key to production of food and energy to satisfy human needs,
including poverty alleviation and healthy humans. As indiscriminate development threatens critical
ecosystem services and biodiversity, the need to account for the environment has emerged as an
important consideration in sustainable water management (Millennium Ecosystem Assessment,
2005). Enabling society to address sustainability challenges, and develop appropriate solutions,
requires an ability to provide reliable predictions of changes to freshwater resources, their
distribution, circulation, and quality under natural and human-induced changes from local to global
scales, including changes that are part of water management (Srinivasan et al., 2017).

We cannot understand, let alone make future predictions of, water resource system dynamics,
without understanding how the issues of economic gain, environmental degradation, and social
inequities play out in society, and how social perceptions of these issues impact management
decisions relating to water consumption, allocation and pricing, human settlements, infrastructure
development, and environmental protection (Blair and Buytaert, 2016; Srinivasan et al., 2016). Such
understanding will remain incomplete until we fully grapple with issues arising from human culture,
including how components of culture – values, beliefs, and norms relate to water uses, livelihood,
and the environment (Sivapalan and Blöschl, 2015). It is increasingly recognized that cultural factors
are likely to influence changes in water management decisions and outcomes (Caldas et al., 2015),
raising questions about what have become 'conventional' assumptions about humans as rational,
utility maximisers who make decisions based upon complete information. Although economic
models of altruism and impure altruism (i.e., "warm glow" effect: caring about others or the next





generation not just out of altruism but because they get pleasure out of it themselves) have been successful in predicting the effect of prevailing values and norms on human behaviour and actions (Andreoni, 1989; Banerjee and Newman, 1993), they remain limited in accounting for the consequences of the human actions on societal values and norms in return.

The inter-disciplinary field of socio-hydrology was launched with the aim of studying the dynamic, two-way feedbacks between water and people in coupled human-water systems. In particular, socio-hydrology (SH) seeks to understand and interpret patterns and phenomena that emerge from two-way feedbacks in coupled human-water systems as a consequence of water management decisions and actions. Indeed, the subject matter of socio-hydrology are the many diverse phenomena that emerge from these two-way feedbacks and manifest as puzzles and paradoxes, exhibiting differences but also similarities between places, and reflecting distinct hydro-climatic, eco-environmental, and socioeconomic backgrounds (Sivapalan et al., 2014). Examples include the agrarian crisis in booming emerging economies such as India (Pande and Savenije, 2016), increasing levee heights in urban environments in spite of increased flood risk (Di Baldassarre et al., 2013) and the peaking in water resource availability in agricultural basins as they undergo development (Kandasamy et al., 2014; Liu et al., 2014).

Several place-based socio-hydrology studies in basins dominated by agricultural development (Tarim: Liu et al. 2014; Murrumbidgee: Elshafei et al. 2014, van Emmerik et al. 2014; Lake Toolbin: Elshafei et al. 2015) have highlighted a shift in water use behavior from an initial focus on agricultural production to an increasing emphasis on environmental conservation, a shift that has been called the pendulum swing (Kandasamy et al., 2014). Socio-hydrology models developed to reproduce these observed dynamics attributed the shift to changing human values and norms, which were tracked indirectly through proxies (e.g., environmental degradation). For example, van Emmerik et al. (2014) modeled the human decision to allocate more or less water to agriculture or to the environment on the strength of a dynamic 'social' state variable called environmental awareness, which reflected societal perceptions of the environmental degradation within the prevailing value systems or culture (see also Di Baldassarre et al. 2013 for awareness of floods in the context of coupled human-flood systems). In the socio-hydrological model of van Emmerik et al. the human response to changing environmental awareness is captured through an appropriate



constitutive relationship, chosen in a somewhat intuitive way. Hence, the parameters governing the
constitutive relationship could only be obtained through calibration of the overall model and would
always be challenged unless they are verified to be right for the right reasons. Prediction-wise, both
in time and space, confidence in such place-based models will be low so long as the constitutive
relationship cannot be independently validated or theoretically justified.

Going forward, there is a need to generalize SH models both for predicting future socio-hydrological
outcomes in one location and/or to apply them at other locations. Case studies have demonstrated an
inherently dynamic quality to changing values and norms in relation to water use or environmental
behaviour, but how to measure or "value" values and norms directly and independently of models
remains as yet unresolved. Even if they can be measured in specific places, we need a broad
theoretical framework that encapsulates the many physical and social controls that govern changing
values and norms in order to synthesize data or measurements from many places across the globe
and develop broad generalizations. These remain major challenges to the progress of socio-
hydrology as the science underpinning sustainable water management (Pande and Sivapalan, 2016)
and thus provide the motivation for this paper. Our aim is to position the progress made by SH
models to date towards incorporating changing values and norms in the context of extant social
science theories, and in doing so, to articulate possible ways forward to make major advances in the
future.

This paper begins with a review of recent place-based, socio-hydrological modelling studies (van
Emmerik et al., 2014; Elshafei et al., 2014, 2015; Roobavannan et al., 2017) that have incorporated
changing values and norms by connecting them to measures of the states of basin economy and/or
environmental health via assumed functional relationships. Next, we draw connections between
extant social theory and recent SH studies that indicate how values and norms influence social
behaviour towards the environment. The paper then outlines challenges and opportunities for
generalising SH models, especially in respect of changing values and norms, so that more reliable
predictions can be made across time and space. This includes a re-calibration exercise to demonstrate
the value of new kinds of social data. This also includes exciting new avenues such as virtual social
experiments or data mined from novel sources such as social surveys and media. It concludes with
the possibility of generalising relationships between changing values and norms and human behavior



in respect of the environment, benefiting from more place based studies. In this way, it underscores
the need for more comparative analyses across many such case studies so that generalised
relationships can be synthesised that are transferrable to ungauged locations.

## 2. VALUES AND NORMS IN SOCIO-HYDROLOGY MODELS

Following Wescoat (2013), the socio-hydrology literature has tended to define values and norms as
the over-arching goals of individuals and of whole societies in respect of water use, conservation,
and sustainability. Prior research in SH has allowed values and norms to undergo dynamic changes.
Sivapalan et al. (2014) proposed a socio-hydrology framework which uses values and norms as
drivers of the decision making that shapes society's goals and actions, and are in turn shaped by the
outcomes for human wellbeing that result from past human decisions (Figure 1). In this way, values
and norms are seen as endogenous to coupled human-water systems, co-evolving with the changing
dynamics of water resource systems (Norton et al., 1998; Sivapalan and Blöschl, 2015). So far in
SH research, values and norms have been lumped together and represented by proxy variables. Next,
we illustrate this through several examples.

### 2.1 Environmental awareness

van Emmerik et al. (2014) developed a SH model of the Murrumbidgee river basin (MRB)
in eastern Australia to explain an observed "pendulum swing", i.e., a shift in water management
focus away from economic development and towards ecosystem health. This shift was hypothesized
to be the outcome of changes in values and norms in the community in respect of economic well
being and ecosystem health. In the model, the dynamics of changing values and norms were
represented by environmental awareness, a proxy state variable that reflected adverse changes to
ecosystem health. It was assumed that environmental degradation occurred when too much water
was extracted for agricultural activities aimed at advancing economic wellbeing of the community.
As a result, less water reached downstream wetlands. When wetland storage became lower than a
specified threshold, ecosystem health suffered noticeably to be felt in the community, which was
then reflected in the environmental awareness. Enhanced environmental awareness then triggered
human action, in the form of reductions in water allocation to agriculture, leading to reductions in
irrigated area, and increased water allocation to the environment. The situation would reverse itself
upon a return of increased downstream environmental flows, restoration of wetland storage and





improvement to ecosystem health.

The representation of environmental awareness in van Emmerik et al. (2014), although simple,
represents a first attempt on the intuitive relationship between values and norms about perceived
threats to ecosystem health and changes to water management actions. Note that other effects or
characteristics of environmental degradation, such as changing water tables, or salinization of the
soil, were not taken into account. Furthermore, regional or national policy is not taken into account
in the formulation of environmental awareness. Finally, the functional form of the equation was
calibrated using data on population, total irrigated area, agricuture water utilisation.

## 2.2    Community sensitivity
Elshafei et al. (2014) expanded further on the intuitive causality between changes to
community values and norms in respect of ecosystem health and consequent water management
actions by humans. They elaborated on how agri-centric values conflicted with environmental
values and influenced water use behavior and proposed a framework that modeled the competition
between economic development and environmental awareness using 'community sensitivity', a
new social state variable. They presented a feedback formulation where water use behavior is
influenced by changing values and norms relating to the environment and economic well-being, as
reflected in the community sensitivity. For the first time the authors brought in broader (e.g.,
regional) climatic, political and socio-economic contextual variables that may influence local
values and norms in respect of water use, e.g., rapidly diversifying economic growth. Elshafei et al.
(2015) explicitly demonstrated that environmental degradation impacted community sensisitivity
and consequently water use behaviours. The foundation of their proposed framework was driven
by the hypothesis that the coupled system dynamics are driven by the competition between a
positive feedback loop (Economic-Population Loop) and a negative feedback loop (Community
Sensitivity Loop).
## 2.3 Economic diversification and institutions:
Roobavannan et al. (2017) presented a rigorous validation of the community sensitivity
concept of Elshafei et al. (2014) and further extended it to account for the relative dependence of the
basin economy on agriculture. Roobavannan et al. (2017) assumed that the tradeoff between
economic wellbeing and environmental health at the community level depends also on contextual



factors such as economic diversification. In this way the resulting SH model was able to explain the
importance of economic diversification and sectoral transformation on the community sensitivity
that then impacted human water management actions.

Roobavannan et al. (2017) also introduced a fish spices richness (FSR) index (Yoshika*wa et al*,
2014) as a separate proxy for ecosystem health. They also used time series of economic development
(measured by total irrigated area and irrigation water utilisation) and diverse proxies for technology
(i.e., patents) and water use behavior (e.g., environmental behavior based on fish species richness
index) in validating the dynamic changes to community sensitivity.
## 3. VALUES, BELIEFS AND NORMS AS DYNAMIC VARIABLES
So far in SH modelling research, aspects of human culture that drive human behaviour in respect
of water management – i.e., values and norms – have been treated in a lumped way, represented by
proxies, in a black-box manner. Moving SH forward requires opening the 'black box' of culture by
questioning the assumptions behind and more clearly measuring and modelling cultural factors. For
example, if values are conceptualized as over-arching goals of society (Wescoat, 2013), are they
individual goals or collective goals associated with the emergent structure of a coupled human-water
system, or both? Similarly, how malleable are values and norms as aspects of a coupled human-
water system? Moreover, under what conditions should values and norms be expected to change, or
remain stable? For that matter, what are the mechanisms through which values and norms might
change, and the human behaviours and actions that result from them?

The ingredients for understanding the role of changing values and norms in coupled human-water
systems can be summarized as (a) forward loop: theories of how individual values influence
individual norms and behavior regarding water use, (b) backward loop: theories of why and how
collective behavior can engender change in individual norms regarding the use of water for
agriculture or the environment, (c) role of institutions in enabling changes in water policy that reflect
collective behavior towards the water environment, (d) data that can provide information on proxy
variables including environment related behavior and patterns and (e) models that use proxy data to
conceptualize processes (a)-(c) in interpreting related patterns. Future work in SH will necessarily



grapple with these types of questions that further elucidate the role of values and norms in coupled
human-water systems.

**3.1 Values, Beliefs, and Norms: VBN theory**
One line of conceptualization seems particularly promising for moving forward socio-
hydrological research. The Values-Beliefs-Norms (VBN) theoretical framework (Stern et al., 1999;
Ives and Kendal, 2014) is grounded firmly in social-psychological theory and has been empirically
tested as a framework for understanding how cultural factors (i.e., values, beliefs and norms) shape
environmental decision-making, and water use behaviour in particular, in a wide array of contexts.
Figure 2 presents a stylized version of a VBN model linking values, beliefs, norms, and behaviours.
In this framework, behaviours are motivated by proximate norms, or obligations to act. Norms
themselves are shaped, or activated by beliefs, including a person's awareness of the consequences
of their actions, how a person ascribes responsibility for their actions etc.  More generally, norms
are shaped by a person's ecological worldview, or how a person views humans vis-à-vis the natural
environment (i.e., are humans *a part of* the natural environment, or *apart from* the natural
environment). Ultimately, the VBN framework posits values-deeply-held, guiding principles about
right and wrong – as the basis of water use behaviour in the context of socio-hydrology. Values are
often assumed to be unchanging, relatively stable, and generally unquestioned principles that
motivate water use behaviour and water policy actions indirectly through beliefs and norms.

The VBN framework is capable of being incorporated into SH models for the purposes of modelling
dynamic feedbacks within the human component of the system or between the human and
environmental components of the system (Caldas et al. 2015).  Incorporating VBN into SH models
requires addressing the questions raised above in greater detail, among others, but especially the
question of where the feedbacks between values, beliefs, norms and behavior occur in the process
of management and the competitive use of water resources.

To illustrate how values, beliefs and norms influence behavior, consider a simplified example of a
farmer of English descent in the MRB who migrated into the basin in the early 1900s and farmed
rice. The behaviour of this farmer towards wetlands is influenced by how the farmer and the farming
community *believe* their water use affects what they hold dear or value. Implicitly, this means that





their behaviour towards the environment depends on how they value water, or what they believe the
water should be used for. These are questions of *values*, and values help navigate decisions that must
be made about trade-offs between different valued end goals, or uses. Here, one key trade-off is
between water for agricultural production (i.e., to support the viability of the farm operation and
farmer's livelihood) and water for the environment (i.e., to support environmental flows,
biodiversity, and ecosystem services). Humans can hold multiple values, and place different
'weights' or emphases on each of the values that affect a particular decision with regards to water
use. The farmer may, for example, make a water use decision by drawing on a combination of self-
interest/egoistic values (e.g., using water to support the economic well-being of their family,
household, and farm), humanist-altruistic values (e.g., conserving water to preserve the long-term
viability of the rural community), and biospheric-altruistic values (e.g., conserving water to preserve
wildlife habitat and ecosystem services). A first step toward modelling this type of VBN process
could be to assign weights for each value, allowing behaviours to change in correspondence to the
weights that each value type exercises over time. Scaling up from the individual-level, value types
can be identified from prevailing complexes of VBN processes in a basin so that SH dynamics in a
basin are outcomes of generalised behaviours emerging from a distribution of basin residents laden
with different value types and complexes. From this perspective, VBN elements at an aggregate
level in a basin can become dynamic. For example, degrading ecosystem functioning, such as the
drying of wetlands, can bring more uncertainty and risk over time to the things the farmer values
(i.e., income, family, farming, community, the environment, etc.) and/or altering the farmer's beliefs
(i.e., worldview, awareness of adverse conseqeunces, or perceived ability to reduce threats to things
of value), shifting their behaviour away from a more egoistic, or agri-centric, orientation and towards
wetland conservation and restoration. This is a very simplified example of a complex set of processes
operating at multiple scales, but it illustrates how values, beliefs, norms, and behaviour might be
seen to co-evolve and change through feedbacks in a coupled SH system.

There remain important gaps in how to identify the requisite components of VBN processes through
measurement, how to scale up these processses from the individual level, and how to model
feedbacks. However, as mentioned before, there has already been progress in this direction in the
SH literature. Place-based SH models (van Emmerik et al., 2014; Roobavannan et al., 2017; and
Elshafei et al., 2014, 2015) have mimicked various regimes that result from a different balance



between economic or agricultural development and environmental health due to changing values,
beliefs and norms. van Emmerik et al. (2014) was able to model the four eras described by
Kandasamy et al. (2014), from an exclusive focus on agriculture, to environmental restoration. A
crucial aspect has been the inclusion of a sub-model to quantify environmental health. The
community sensitivity framework of Elshafei et al. (2014) was applied to two Australian catchments,
and in both cases, different regimes could also be distinguished. Interestingly, the inclusion of human
feedback that integrates a variety of influences as a response to changes in ecosystem health was
done in a completely different fashion. In van Emmerik et al. (2014) a simple memory function
governed by wetland storage sufficed, whereas in Elshafei et al. (2014) more complex community
sensitivity equations were introduced, both linking water use related beliefs and behavior through
bi-directional feedbacks. Roobavannan et al. (2017), advanced this a step further by representing
community level belief about the environment, i.e., environment sensitivity, as a consequence of the
distribution of weights that individuals attach to enviro-centric versus anthropo-centric values. Such
a distribution was made contextual, i.e., it depended on economic diversification. The endogenous
treatment of values and norms by these studies (van Emmerik et al., 2014; Elshafei et al., 2015;
Roobavannan et al., 2017) have implicitly followed the general logic of elements of the VBN theory
presented above, even if this was originally unintended (see Figure 2) , and have therefore responded
to the challenges of incorporating feedbacks from water use behavior to beliefs and water
management norms, consistent with the notion of endogenous and dynamic culture of Caldas et al.

294  (2015).

**3.2 Validation of Modeled Changing Values and Norms**
Place-based SH models have relied on proxy measures such as environmental degradation to capture
changing values, beliefs, norms and behaviors and their parameters were obtained by calibration.
Despite the advantages of this approach, confidence in these models remains low, as the models
struggle to be indepedently validated. To address the validation challenges faced to date in model-
based socio-hydrology case studies, Elshafei et al. (2015) proposed that socio-centric approaches
(such as newspaper content analysis) be employed to assess evolving community sentiment over
long time periods.

Along these lines, Wei et al. (2017) recently analyzed the content of newspaper articles to measure





and quantify the evolution of societal values and preferences in relation to water management issues in Australia over a 169 year period. The results of Wei et al. (2017) are especially informative to the growing body of socio-hydrology literature focused on Australian study sites, in particular the Murray Darling Basin (MDB). Their findings support the hypotheses put forward in Kandasamy et al. (2014) and Elshafei et al. (2014), both of which postulate a shift in societal values from an anthropo-centric to an enviro-centric focus over time.

The work of Wei et al. (2017) thus signals an important step forward for the socio-hydrology research community as its results demonstrate how an autonomous socio-centric analysis method may be employed to provide independent validation for conceptual theories and coupled modelling approaches carried out within the same broad geographical region. This more complete analysis of societal values now enables us to go back and compare the results of this independent study against the predictions made by previous SH models. More specifically, Wei et al.'s (2017) results corroborate Kandasamy et al.'s (2014) proposed pendulum swing in societal sentiment in the Murrumbidgee Basin over a century timescale. As can be seen in Figure 3, observed (Figure 3a, Kandasamy et al., 2014) and modeled (Figure 3b, van Emmerik et al., 2014) time series of economic development (proxied by total irrigated area and irrigation water utilisation) correspond with the evolution of societal sentiment shown in the bottom panel of Wei et al.'s (2017) results (Figure 3c). Moreover, the narrative for each of the three phases described in Wei et al. (2017) repeats the timing and spirit of the phases depicted in Kandasamy et al. (2014), van Emmerik et al. (2014) and Elshafei et al. (2014, 2015) (Figure 3).

Another important implication of Wei et al.'s (2017) results in relation to Elshafei et al.'s (2014) proposed conceptual SH model is that they provide strong support for theories underpinning the use of the composite 'community sensitivity' variable put forward therein. Figure 4a,b illustrates that when societal values are initially focused on economic development the change in the community sensitivity variable (dV/dt) trends negative (i.e., society is predisposed towards anthropo-centric behaviours), whereas as societal values evolve towards environmental sustainability the change in community sensitivity variable trends positive (indicating a behavioural tendency towards conservation). Wei et al.'s (2017) findings thus provide strong validation for the non-linear dynamics observed in previously published coupled SH models that adopted alternate proxies for modelling



the change in societal values and norms in relation to water resource management over time (i.e.,
Elshafei et al.'s (2014, 2015) composite community sensitivity variable and van Emmerik et al.'s
(2014) environmental awareness variable).

It is worth noting that Wei et al.'s (2017) results are not particular to a specific basin, but rather are
intended to reflect a broader national or regional view. Validated SH models that endogenized water
related beliefs and norms are distinct from regression based models that are not causal (e.g., Wei et
al., 2017). The in-built non-linear dynamics allow possible 'extrapolation' of the coupled human-
water dynamics across a gradient of hydro-climates, societies and economies, although this requires
more work and testing. Similar to regionalisation techniques in hydrological modeling, socio-
hydrological regionalisation will mean how the parameters of the coupled SH model, such as
curvature parameter of the distribution function that trades off enviro-centric values with anthropo-
centric values (Roobavannan et al., 2017), vary across different societies. Regression based models
cannot be extrapolated to another place or time as there are no causal linkages provided to explain
the transitional shifts in societal values observed therein. In other words, regression models that do
not internalize coupled human water system dynamics can at best be used for 'interpolation' (i.e.,
can only explain the dynamics within the domain of the data) or data analysis. Nonetheless,
verification of coupled models with data such as those presented in Wei et al. (2017) is important as
it enables the discovery of fundamental principles of human behaviour through the validation of
internal dynamics within the coupled models, and ultimately aids in the generalisation of socio-
hydrologic system dynamics. The following shows how newspaper content analysis effectively plays
the same informative role as Fish Species Richness, i.e., FSR (i.e., proxy for condition of ecology),
in modelling water related endogenous behaviour.

In order to illustrate how newspaper content analysis serves as a complementary source of
information that can be used in socio-hydrological modelling, the Wei et al. (2017) data was used to
re-calibrate the 'environment awareness' state variable of van Emmerik et al. (2014). Instead of
wetland storage which was used in van Emmerik et al. (2014), the Fish Species Richness ($r$) is now
used as a proxy of environment health. The temporal dynamics of environment awareness (E) is
assumed to be given by the following differential equation (van Emmerik et al., 2014):





$$\frac{dE}{dt} = \varepsilon(r)$$

where $\varepsilon(r)$ is the rate of accumulation/depletion of environmental awareness, which is a function of

$r$. The functional form of $\varepsilon(r)$ is assumed to be given by:

$$\varepsilon(r) = \begin{cases} \alpha\,[\exp(\beta r) - 1], & r < r_c \\ -\lambda, & r > r_c \end{cases}$$

where $r_c$ is the critical Fish Species Richness below which environment awareness is expected to

increase exponentially governed by parameters $\alpha$ and $\beta$ and $\lambda$ is dissipation rate of environmental

awareness when the ecosystem is healthy, i.e., $r > r_c$. The Fish Species Index, $r$, (Yoshika*wa et al*,

2014) is estimated by the following power law function:

$$r = \beta_0 Q_B{}^{\beta_1}$$

where $Q_B$ is the flow in the downstream streamflow (i.e., environmental flow) and $\beta_0$ and $\beta_1$ are

parameters of the FSR index (Yoshikawa *et al.*, 2014).

Values of the parameters $\alpha, \beta, \lambda, r_c$ need to be calibrated. In the absence of social data to calibrate

the model, van Emmerik et al. (2014) used other basin-wide hydrological data to calibrate the model.

Here we use the Wei et al. (2017) data to calibrate the model parameters, through application of the

GLUE method. Initial estimates for the parameters are obtained manually making sure essential

dynamics are captured. After that, 100,000 random samples of parameters (uniform sampling) that

lie within the range of 50% to 150% of the initial values are obtained.

Figure 5a shows the modeled environmental awareness by van Emmerik et al. (2014) and a

comparison with that calibrated to the Wei et al. (2017) data (Figure 5b). The environmental

awarness (E, Figure 5a) bears a remarkable similarity to that obtained by Wei et al. (2017) through

newspaper content analysis. Even though van Emmerik et al. (2014) at that time was not privy to the

Wei et al. (2017) data, the model already succeeded in capturing the change in community's values

and norms regarding water resources. While naturally attracting criticism for the lack of direct

calibration, in hindsight the validity of the approach may now be appreciated and that new social

data such as Wei et al.'s (2017) can be used to validate predictions of changing values and norms.

Figure 5b shows how with foresight and with availibity of complementary societal values data of

Wei et al. (2017) (see dashed line), the FSR can robustly simulate E. In doing so it provides



independent validation of the model results of van Emmerik et.al. (2014) and the approch that was adopted at the time.

## 4. FROM PLACE-BASED TO GENERALIZED MODELS: CHALLENGES AND OPPORTUNITIES

The pathway to generalisation of SH models is an important goal that allows future prediction (extrapolation in time) and translation of SH models at other geographical locations (extrapolation in space). It provides an important means for the adoption of socio-hydrology in the practice of long-term or strategic water resource management. Generalisation needs to address both the proxies used in SH modelling and the data used to calibrate them, as recent SH modelling studies have highlighted.

Models provide languages or templates in terms of which the following three aspects can be interpreted: 1) how beliefs and norms depend on values, 2) how values and norms influence individual behavior towards the environment, e.g., the wetland health or releasing environment water for bio-diversity, and 3) how pro-environmental behaviour of some in the community (e.g., rallies by the Green Movement) can influence the beliefs of others in the basin and bring about a change in water management (i.e., the feedback). Such templates also enlighten us with variables that need to be measured, so that multiple concepts via the models can be tested and can improve our system understanding.

For example, the policy change in the 1990s in MRB led to increased environmental flow. To interpret this in terms of change in water management norms of the MRB, models need to link beliefs and norms to water use behaviour within the basin. This needs information on a range of relevant values such as altruistic values (i.e., healthy MRB for present and future generations, enough money for the next generation) and egoistic values (i.e., making money), along with information on beliefs, norms, and behaviours, such as how water is being used.

### 4.1 Measurement of changing norms and values

Direct measurement of social value is often very difficult, resulting in the use of indirect methods (or proxies). Studies have attempted to understand social values on pro-environmentalism (Bengston





1994; Ives and Kendal 2013) and could be differentiated based on the method of measurement. Assigned values can be expressed in either monetary or non-monetary terms, and are relevant to economic and psychology approaches. In a social science context, assigned values have been quantitatively measured using a variety of techniques, including survey and interview approaches with the help of psychometric scales used in psychology (Bengston, 1994), social experiments in behavioral economics (Janssen et al., 2014; Yu et al., 2016) and content analysis (Seymour et al., 2010; Bark et al. 2016a; Xu and Bengston 1997; Wei et al., 2017). Economic valuation offers another set of useful approaches to inform natural resource management (Farber et al., 2002; Pande et al., 2011; Loomis et al., 2000; Norton and Noonan, 2007; Wilson et al., 1999; Bark et al., 2016b). Non-market valuation (Smith, 1993), contingent valuation (Bateman et al., 2006 ) and other related techniques have been extensively used over the decades and enabled the exploration of how people 'trade-off' their environmental values in decision-making (Freeman 1993). This enables (i) values to be measured for large and diverse groups of people, (ii) changes in values to be tracked across groups of people or across time, and (iii) models to be developed to predict values based on other factors (e.g., demographics, cultural background).

It is less challenging to observe contemporary water-related behaviour. However, as the time scale of analysis expands, the task of measuring behaviour becomes equally challenging. Paleoclimate proxies such as $\delta^{18}O$ or tree rings,, have been extensively used to interpret water availability as well as social organization in the past (Pande and Ertsen, 2014; Staubwasser et al., 2003). These observations can be supplemented by other forms of indirect measurement of water related behaviour such as newspaper content analysis, records of memberships in activist organisations, and can strengthen proxy observations of pro-environmental behaviour in the near past.

### 4.2 Utilisation of new types of data

A challenge related to model transferability is generic data needs. If community sensitivity functions of van Emmerik et al. (2014), Elshafei et al. (2015) and Roobavannan et al. (2017) are able to assess some trade-off between enviro-centric and anthropo-centric values types, global socio-economic data sets such as the World Value Surveys (WVS, 2017) and UN demographic datasets (UN, 2017) might offer the possibilty of quantifying values, so that models can be transferred to unmonitored locations. Whether such data sources can be used to quantify such values remains a very important



open question.

In the past, the use of soft data in hydrological modelling has been demonstrated to provide
additional insights into the functioning of ungauged basins, and has in some cases been used to
successfully assess the realism of a model (see e.g., van Emmerik et al., 2015). Similarly, socio-
hydrological systems face similar problems of extrapolation to other places, as numerical data series
do not always exist to calibrate or validate SH models. Wei et al.'s (2017) use of newspaper content
data to compute a numerical expression of environmental sustainability and economic development
demonstrates the benefits of further exploration of this type of new data sources since it can allow
the calibration of SH models, as shown in Figure 5. Future efforts should therefore not only be
limited to developing new SH modeling frameworks, but also entail finding new ways to access
information and translate it into numerical expressions, e.g., indices such as FSR, that can be used
for model validation, and model realism assessment.

A new era of data-driven science (Peters-Lidard et al., 2017) is dawning, with increased
computational power, new proxies and alternative data sources. Smart distillation of information
from alternative sources (e.g., web databases, social data, other types of Big Data) may provide the
valuable auxiliary data required to take the next step in SH model development and provide an
innovative way to find and quantify the social proxies which are currently difficult to justify. This
will need to be combined with online data monitoring such as smart sensing and citizen science
monitoring as well as field campaigns to validate model results as well as to obtain socio-
hydrological data relating to e.g. environmental sentiment, local sociatel values, and fertility
conditions. In the future, socio-hydrologists could exploit or mine data/information from such varied
sources, leading to the inclusion of Big Data science in socio-hydrology. This new paradigm
represents a clear set of opportunities for data-mining and data-driven modelling methods in socio-
hydrology. These apply machine learning and 'computationally intelligent' algorithms to elicit,
characterise, quantify and model the myriad, implicit structures and relationships embedded within
complex, multivariate datasets. In so doing, they offer a pathway for formulating new understandings
of the saliency and power of socio-hydrologic variables, and the inter-relationships and behaviours
that exist between them (Mount et al., 2016).



### 4.3 Comparative socio-hydrology studies

Parameters are used to calibrate the proxies to fit local basin data. Comparative studies from several basins will enable better interpretation of what model parameters mean and their character. For example, Roobavannan et al. (2017)'s model of endogenous behaviour could be made more socio-hydrologically meaningful. Its attractiveness parameter relates migration to the difference in unemployment within and outside the basin. A more meaningful representation of this variable, for example in terms of the cost of migration, such as moving costs and the cost of obtaining new skills away from water based employment, will enable regionalisation of associated parameter values and the transfer of models from data intensive basins such as the MRB to data scarce basins such as the Aral Sea.

Comparative studies can also provide the data to develop regional relationships for SH model parameters. Similar to regionalisation techniques in hydrological modeling (Asong et al., 2015; Buytaert and Beven, 2009; Götzinger and Bárdossy, 2007; Merz and Blöschl, 2004; Yadav et al., 2007; Blöschl et al., 2013), socio-hydrological regionalisation will define how the parameters of the coupled SH model vary with different societies and basins. Once defined, regional curves may be used to interpolate parameters and hence models to ungauged locations. Initial efforts have already been attempted in Elshafei et al., (2016) but these need to be improved and validated through more independent comparative studies. Yet another possibility can be of investigating a Budyko type curve for coupled human-water systems with endogenous values and norms that will enable extrapolation of emergent behaviours in space and time. Comparative assessement will also put to test theories, such as those that propose values and norms as emergent properties of a coupled human-water system, such that all its biological constituents including humans and vegetation obey certain metabolic scaling laws (Fischer-Kowalski, 1998; Silva et al., 2006).

In this regard, a new working group on comparative socio-hydrology within Panta Rhei has been launched to serve this purpose (Fuqiang Tian, personal communication). It plans to obtain socio-hydrological data from diverse river basins such as Tarim, Murrumbidgee and Kissimmee, including historical documentation of the evolution of coupled human-water system to develop a generalized understanding of coupled human-water behavior. This is being done through comparative analysis to identify and interpret diverse emergent behavior such as farmer suicides in less developed and



developed countries such as India and Australia respectively, "pendulum swing" in basins in China,
USA and Australia and the levee effect versus memory effect in flood plains across the globe. Such
comparative analyses can prove to be very constructive in identifying general principles that govern
dynamic changes in values and norms.
**5. CONCLUSIONS**
Recent socio-hydrological studies (van Emmerik et al, 2014; Elshafiei et al., 2015; Roobavannan et
al., 2017) in Austrialia have moved closer toward integration with key social science theories of
perception and behavior, and have taken a key step toward endogenizing values and norms. These
models are internally consistent with patterns observed with proxy data of environmental awareness
and water policy change, such as the newpaper articles based proxies of Wei et al. (2017). However,
such theoretically and empirically consistent models are only the beginning of the way forward to
generalizing models and its predictions for sustainable water management.

Human culture – comprised of values, beliefs, and norms – is key to understanding stability and
change in coupled human-water systems. Often, such variables and related closure relationships
within socio-hydrological models are latent and hard to observe. This poses challenges in testing
and confirming the realism of assumed relationships. However, with the advent of the information
intense era, diverse proxy data sources such as citizen science observatories, and social media can
be harnessed and novel big data algorithms can be used to process them in a form that can be of use
to socio-hydrological models.

Yet such opportunities can only build confidence in our place based understanding of a socio-
hydrological phenomenon such as the pendulum swing observed in the Murrumbidgee River Basin.
What we need are generalized relationships or principles underlying emergent phenomena if we are
to stand up to the challenge of making predictions in ungauged locations in space and time. This
clearly calls for more place based studies, both past and present and across spatio-temporal scales,
that are backed up by novel socio-hydrological observations such as historical accounts and socio-
centric data, and a comparative analysis of such studies where similar emergent phenomenon has
been observed to help synthesize the underlying socio-hydrological principles.





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





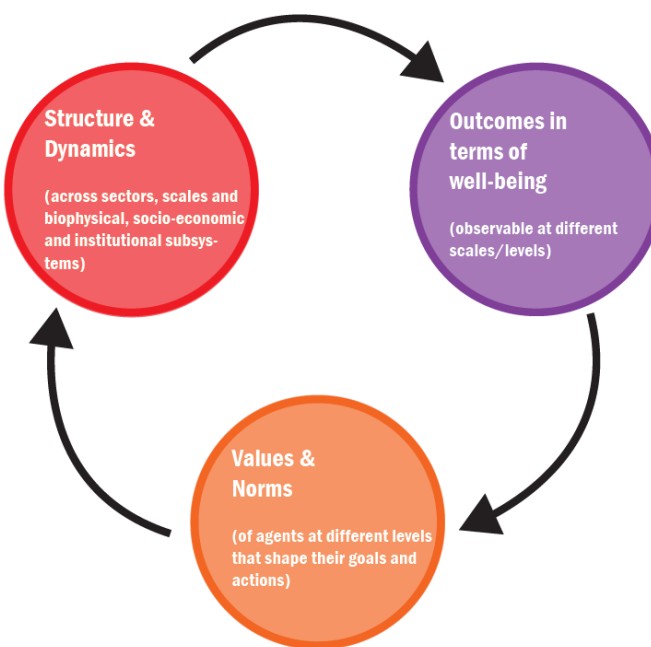



Figure 1: Framework proposed by Sivapalan et al., (2014). Socio-hydrology models use proxies for
environment degradation and for economic well being














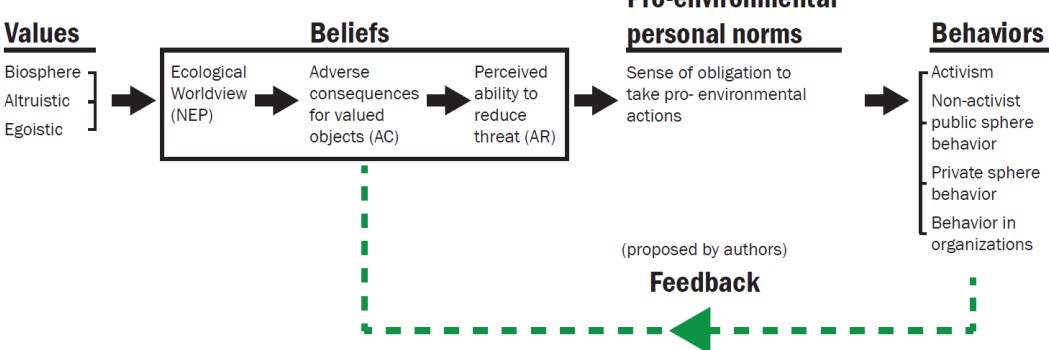


Figure 2: Value Belief Norm (VBN) theory. Adapted from (Ives and Kendal, 2014; Stern, 2000).
The feedback (green arrow) from communal behavior to individual beliefs is introduced here by the
authors to recognize that it has indeed been included in recent SH studies in preliminary ways and
(van Emmerik et al, 2014; Elshafei et al, 2014; Roobavaanan et al, 2017) needs to be formalized in
future studies.



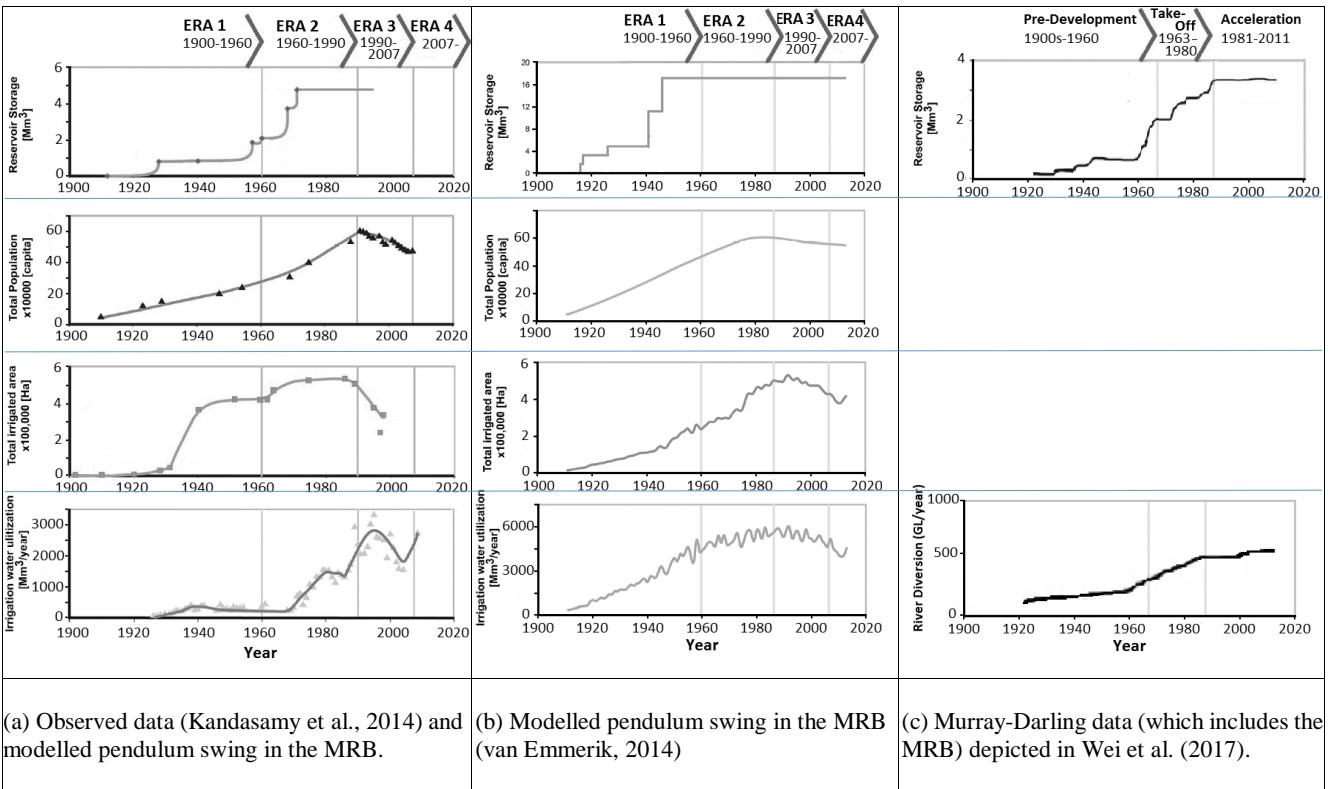

(a) Observed data (Kandasamy et al., 2014) and modelled pendulum swing in the MRB.

(b) Modelled pendulum swing in the MRB (van Emmerik, 2014)

(c) Murray-Darling data (which includes the MRB) depicted in Wei et al. (2017).

**Figure 3.** Observed and modelled pendulum swing in the MRB during the period 1910–2013. Era 1 (1900-1980) Expansion of agriculture and associated infrastructure, Era 2 (1960-1990) Onset of environmental degradation, Era 3 (1990-2007) Establishment of widespread environmental degradation, Era 4 (207-2014) Remediation and emergence of environmental customer. The eras correspond to phases in Elshafei et al (2015): Expansion (1911-1960), aggressive rate of expansion and active modification of water balance; Contraction (1960s), plateau in anthropogenic modification; Recession (1970-2002), cumulative negative impacts on economic and



environmental well-being; Recovery and new equilibrium (2002-present), Adoption of remedial measures; and in Wei et al. (2017):
Pre-development (1900s-1960s) Societal values dominated by economic development; Take-off (1963-1980) Societal values reflected
increasing environmental awareness due to outbreak of pollution events; Acceleration (1981-2011) Growing shift in societal values
towards environmental sustainability.



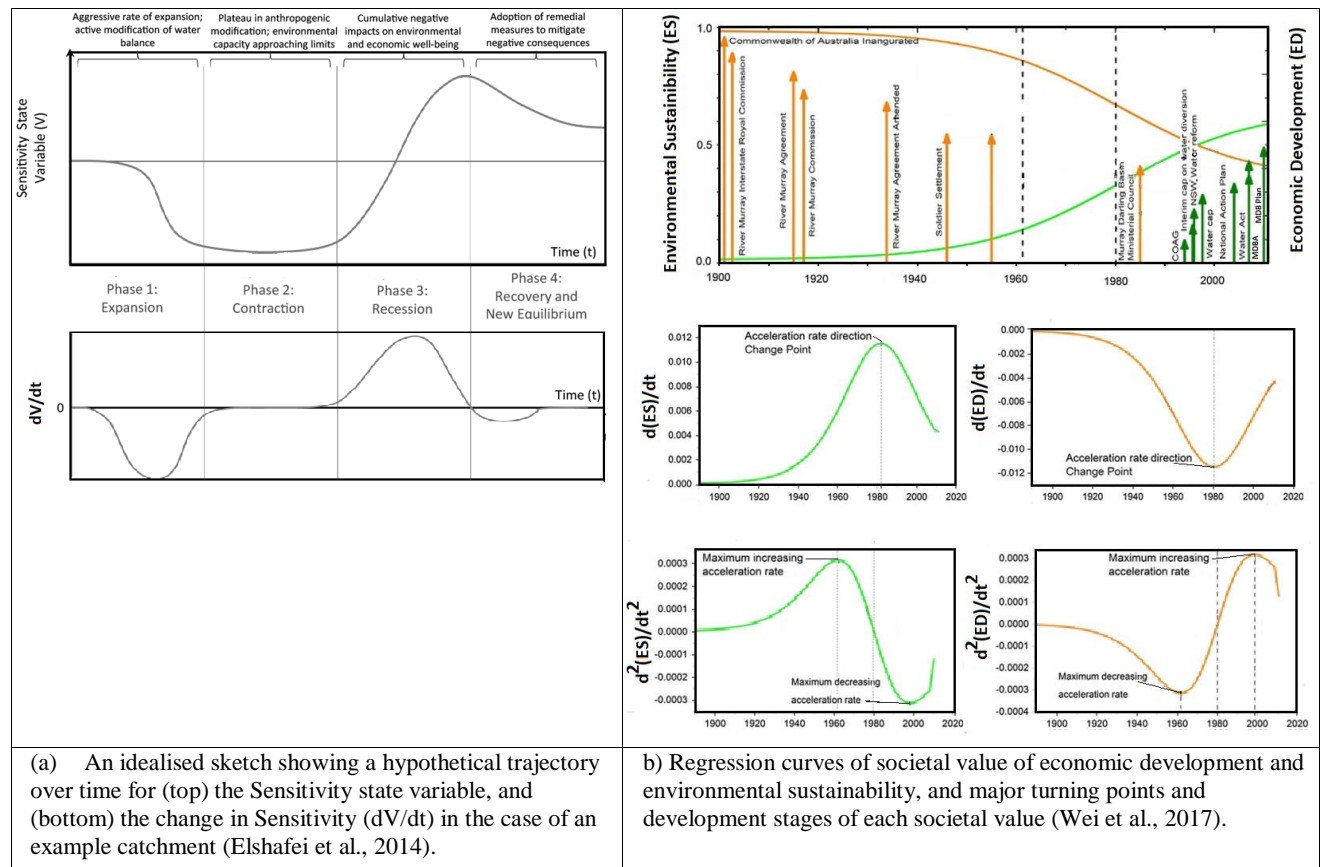

| (a) An idealised sketch showing a hypothetical trajectory over time for (top) the Sensitivity state variable, and (bottom) the change in Sensitivity (dV/dt) in the case of an example catchment (Elshafei et al., 2014). | b) Regression curves of societal value of economic development and environmental sustainability, and major turning points and development stages of each societal value (Wei et al., 2017). |
|---|---|

**Figure 4.** Defining shifts and turning points of stages of societal values




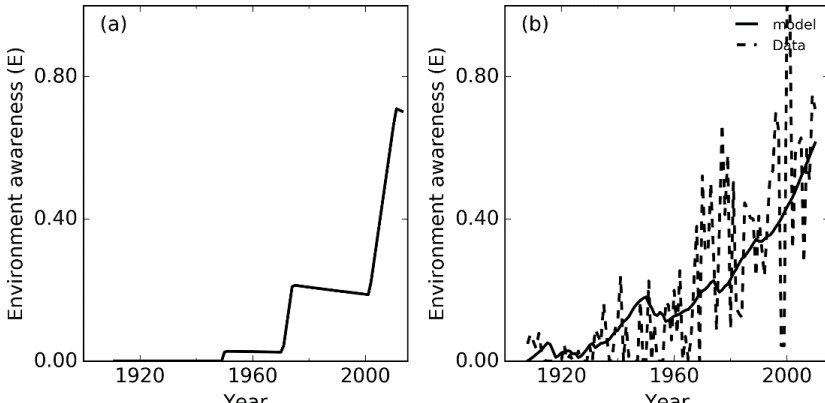



**Figure 5.** (a) Variation of modelled environment awareness by van Emmerik et.al, (2014) using calibrated model with hydrological and population data (b) variation of modelled environment awareness using calibrated model (solid line) with societal value data (data from Wei et al. (2017), dashed line) of water resources for environment stability.

727

728

729