# Peer review of "NORMS AND VALUES IN SOCIO-HYDROLOGICAL MODELS"

_Hydrology and Earth System Sciences, 2017_

## Referee Comment (RC1) · Y. Wei (Referee) · 1 Oct 2017

I enjoy reading this manuscript. A landscape at water catchment is a holistic system in which nature and culture co-evolve. This begs the question: to what degree did the cultural construct influence the water catchment hydrology, and vice versa? However, the cultural construct (societal values) has not been adequately studied in existing hydrological models, except those studies mentioned in the manuscript. Therefore, this review is important by bringing this knowledge gap to the hydrology community (HESS). I would like to recommend this manuscript to be accepted, subject to responses to the comments as follows:

Culture is a notoriously slippery concept, has no agreed-upon definition across social

science fields. There are more 170 definitions of 'culture' in the literature. Culture is often perceived to be opposed to nature, becomes synonymous with civilization. Culture is defined operationally as a set of common values, norms and attitudes shared by the majority of a region population, which is arguably the most important mediating mechanism that links us not only with other human beings, but also with the rest of nature of which we are part and within which we live (Keesing 1974). To talk about cultural change is one thing. To measure them precisely is quite another. The study of cultural evolution has traditionally been the purview of anthropology and sociology. Past attempts to explain cultural evolution used the 'thick description' rather than explanatory approach which would not distinguish between explanandum and explanans. It is known that they have poor predictability. This is why culture (societal value) has not been nicely integrated in the hydrological models. However, these disciplinary studies provide the fundamental basis for any attempts of quantifying the societal value. So, I would like to this manuscript to include a more thorough review of measurement and explanation of societal value in these disciplines.

VBN is one of many theoretical frameworks in sociology which explains the impact of the value-belief-norm on individual or societal decision-making and practice. However, I do not think it is practical in the context of socio-hydrology, in particular when we aim to simulate and reconstruct the historical societal value. Given the limited documents (data) sources, how can you obtain data on value, belief and norms? You make detailed difference between value, belief and norms in Figure 2, but you did not make clear difference between these three concepts in text. So I suggest to combine 3.1 and 3.2 and use a general concept to explain the feedbacks between value and behaviour. You did not give a full explanation of Figure 2, and you did not use main info in Figure 2 in your manuscript either, so I would suggest you delete it.

There is a bit repetition between Section 1, Section 2 and Section 4.

Besides our findings in Australia (Wei et al., 2017) which you cited and used the data from, we had published similar findings in China (Xiong et al., 2016). I list it here for

your information. Yonglan Xiong, Zhiqiang Zhang, and Yongping Wei. 2016. Evolution of China's water issue framed in Chinese mainstream media. AMBIO 45 (2): 241-251DOI: 10.1007/s13280-015-0716-y.

Yongping Wei

Please also note the supplement to this comment:
https://www.hydrol-earth-syst-sci-discuss.net/hess-2017-432/hess-2017-432-RC1-supplement.pdf

---

## Referee Comment (RC2) · X. Chen (Referee) · 5 Oct 2017

This paper did a review of socio-hydrology (SH) modeling with a focus on several place-based studies. Based on the review, the authors pointed out the importance of social norms and values in SH models. At the end, the paper proposed potential future pathways of SH models and discussed the challenges to generalize SH models. The manuscript is well written and it is on a topic of interest to the HESS journal audiences. I have the following comments that I hope the authors could address in the revision.

Specific comments:

1. The paper explains the review case studies in multiple sessions with too much details. The focus of the paper should be the knowledge generated from those case

studies. Maybe the authors can find a way to generalize the information provided by these studies.

2. In section 2.2, maybe the authors should add the following reference, since this study is also using the idea of community sensitivity to do SH modeling.

Chen, X., D. Wang, F. Tian, and M. Sivapalan (2016), From channelization to restoration: Sociohydrologic modeling with changing community preferences in the Kissimmee River Basin, Florida, Water Resour. Res., 52, doi:10.1002/2015WR018194.

3. Section 2.3: Roobavannan et al. (2017) is still in review, so it is hard to assess the review materials in this manuscript.

4. Line 288-294: The paper suggested that environment awareness and community sensitivity are both following the general logic of the VBN theory. So maybe the authors can unify the norm/value parameters to one and provide a clear definition based on the VBN theory.

5. Line 448-453: van Emmerik et al. (2014) uses environment awareness, not community sensitivity.

6. Line 511-513: For the three listed river basins, please add the countries they are located in.

7. Line 521: Typo: "Elshafiei". These three references have been repetitively mentioned in this manuscript over 10 times. I think the focus of the paper should be the scientific knowledge that can push SH modeling forward, not the three case studies.

8. Figure 5: The paper spends a fair amount of paragraphs to talk about the parameter "community sensitivity", but the analysis provided by the study is using "environment awareness", which I believe is a different parameter. Following my previous comment, maybe the authors should add explanations about the differences between these two parameters and try to generalize the parameters, which would be a part of the SH generalization process.

Xi Chen

---

## Author Comment (AC2) · 30 Oct 2017

1) Reviewer Comment: This paper did a review of socio-hydrology (SH) modeling with a focus on several place based studies. Based on the review, the Authors pointed out the importance of social norms and values in SH models. At the end, the paper proposed potential future pathways of SH models and discussed the challenges to generalize SH models. I have the following comments that I hope the authors could address in the revision.

Authors response: We thank Xi Chen for his review. We will address all the specific comments below.

Specific comments:

[Figure]

2) Reviewer Comment: The paper explains the review case studies in multiple sessions with too much details. The focus of the paper should be the knowledge generated from those case studies. Maybe the authors can find a way to generalize the information provided by these studies.

Authors response: Thank you for your concern. We will follow your advice and add a conceptual figure, and a one paragraph synthesis along with it, towards the end of section 4 to generalize the information presented in the section. Please note however that section 4 itself was designed to be a synthesis.

3) Reviewer Comment: In section 2.2, maybe the authors should add the following reference, since this study is also using the idea of community sensitivity to do SH modeling. Chen, X., D. Wang, F. Tian, and M. Sivapalan (2016), From channelization to restoration: Socio hydrologic modeling with changing community preferences in the Kissimmee River Basin, Florida, Water Resour. Res., 52, doi:10.1002/2015WR018194.

Authors response: This was an oversight. We have now added the reference.

4) Reviewer Comment: Section 2.3: Roobavannan et al. (2017) is still in review, so it is hard to assess the review materials in this manuscript.

Authors response: Roobavannan et al. (2017) is now published and is accessible via http://www.doi.org/10.1002/2017WR020671. We have also updated the citation in the reference list.

5) Reviewer Comment: Line 288-294: The paper suggested that environment awareness and community sensitivity are both following the general logic of the VBN theory. So maybe the authors can unify the norm/value parameters to one and provide a clear definition based on the VBN theory.

Authors response: The purpose of community sensitivity and environmental awareness variable is to capture the society's changing value and norms and follow the principles

of VBN theory. It should be noted these variables include the value, beliefs and norms together. We will revise the paper to unify the relevant terms and use variables. We agree that we need to further differentiate the variables as we begin to reliably observe them.

6)Reviewer Comment: Line 448-453: van Emmerik et al. (2014) uses environment awareness, not community sensitivity.

Authors response: Thank you for correction. It is corrected.

7) Reviewer Comment: Line 511-513: For the three listed river basins, please add the countries they are located in.

Authors response: Country of respective river basins will be added.

8) Reviewer Comment: Line 521: Typo: "Elshafiei". These three references have been repetitively mentioned in this manuscript over 10 times. I think the focus of the paper should be the scientific knowledge that can push SH modeling forward, not the three case studies.

Authors response: Typo is corrected. We agree with the reviewer that this section may give the impression that we are just repeating three different case studies. Our intention was really to connect them to VBN theory. We will do a better job in the revisions, and thus minimize the apparent repetitions.

Indeed, the focus of this section was to highlight the need to include changing values and norms of society in order to predict future projections. Through this review we explain that recent SH model studies have moved closer toward integration with key social science theories of perception and behavior, and have taken steps toward endogenizing values and norms. We intended to show that these models are internally consistent with patterns observed with proxy data of environmental awareness and water policy change, such as the newspaper article-based proxies of Wei et al. (2017). However, such proxy-reliant models are only the beginning of the way towards

generalized models and their use in predictions for sustainable water management.

9) Reviewer Comment: Figure 5: The paper spends a fair amount of paragraphs to talk about the parameter "community sensitivity", but the analysis provided by the study is using "environment awareness", which I believe is a different parameter. Following my previous comment, maybe the authors should add explanations about the differences between these two parameters and try to generalize the parameters, which would be a part of the SH generalization process.

Authors response: Community sensitivity and environmental awareness are variables defined to capture the changing values and norms in different socio-hydrological models. Community sensitivity is an advance over the previously defined environmental awareness. We agree that they are different in the way they are defined, yet both intend to capture the same concept of changing values and norms in corresponding socio-hydrological models. We however will add text on the difference between their definitions as the referee suggests, that community sensitivity is a more complex description of environment awareness. Both are modeled as memory variables. But while the time scale of the memory of past environmental disaster is kept constant in the case of latter, it is dynamic in the case of latter and depends on community norms in context of its water environment.

Please also note the supplement to this comment:
https://www.hydrol-earth-syst-sci-discuss.net/hess-2017-432/hess-2017-432-AC2-supplement.pdf

---

## Author Response (AR1)

**RESPONSE TO REVIEWERS**

Changes made in response to reviewer comment appears in yellow highlight in the revised manuscript.

**Reviewer #1 Yongping Wei**

**1) Reviewer Comment:**
I enjoy reading this manuscript. A landscape at water catchment is a holistic system in which nature and culture co-evolve. This begs the question: to what degree did the cultural construct influence the water catchment hydrology, and vice versa? However, the cultural construct (societal values) has not been adequately studied in existing hydrological models, except those studies mentioned in the manuscript. Therefore, this review is important by bringing this knowledge gap to the hydrology community (HESS). I would like to recommend this manuscript to be accepted, subject to responses to the comments as follows:
**Authors response:** We thank for Yongping Wei for her positive review. We firmly agree that the degree to which cultural constructs influence catchment hydrology and vice versa remains to be explored in depth.

**2) Reviewer Comment:**
Culture is a notoriously slippery concept, has no agreed-upon definition across social science fields. There are more 170 definitions of 'culture' in the literature. Culture is often perceived to be opposed to nature, becomes synonymous with civilization. Culture is defined operationally as a set of common values, norms and attitudes shared by the majority of a region population, which is arguably the most important mediating mechanism that links us not only with other human beings, but also with the rest of nature of which we are part and within which we live (Keesing 1974). To talk about cultural change is one thing. To measure them precisely is quite another. The study of cultural evolution has traditionally been the purview of anthropology and sociology.

Past attempts to explain cultural evolution used the 'thick description' rather than explanatory approach which would not distinguish between explanandum and explanans. It is known that they have poor predictability. This is why culture (societal value) has not been nicely integrated in the hydrological models. However, these disciplinary studies provide the fundamental basis for any attempts of quantifying the societal value. So, I would like to this manuscript to include a more thorough review of measurement and explanation of societal value in these disciplines.
**Authors response:** We agree that culture can be a nebulous concept and that there are numerous definitions. There are challenges in incorporating culture into socio-hydrological modeling. This is why we explicitly selected the VBN framework, which allows us to identify culture as a property that emerges from the feedbacks between values, norms, and the hydrological system. This is one of the first steps to integrate social science theories linked with values and norms in context of socio-hydrology. Please note that this is an opinion paper on values and norms in socio-hydrological models, which we agree should build upon strong knowledge of the subject matter. For this reason, we have provided a review of VBN theory, which we believe is very well aligned with the current state of the art in socio-hydrological modeling. With further progress in socio-hydrology, we should be able to define the components of culture (i.e., value, beliefs, norms) related to water management and seek the data sources to be exploited. Nonetheless, in the revised paper, we provided an additional review on the measurement and explanation of different values of society in Section 4.1 while keeping to the scope of the paper. Please see lines 442-450 in the revised manuscript.

**3) Reviewer Comment:** VBN is one of many theoretical frameworks in sociology which explains the impact of the value-belief-norm on individual or societal decision-making and practice. However, I do not think it is practical in the context of socio-hydrology, in particular when we aim to simulate and reconstruct the historical societal value. Given the limited documents (data) sources, how can you obtain data on value, belief and norms?

**Authors Response:** Please see our response to the reviewer comment 2. The VBN framework provides us a fundamental basis not only to quantify values but also to quantify the interlinkages between values and norms via beliefs (see Figure 2), norms and human actions via behavior, and human actions and norms via beliefs. Indeed we agree that the complexity of system concepts needs to sacrificed in favor or simpler ones (while maintaining theoretical integrity), such as only piggybacking on feedbacks between values, behavior and hydrological response, according to data availability on values, beliefs and norms (see e.g. Roobavannan et al, 2017). The data challenges are discussed in Sections 4.1 and 4.2.

**4) Reviewer Comment:** You make detailed difference between value, belief and norms in Figure 2, but you did not make clear difference between these three concepts in text. So I suggest to combine 3.1 and 3.2 and use a general concept to explain the feedbacks between value and behaviour.

**Authors Response**: We have provided more detailed discussions of these concepts, provided more detailed definitions and adapted our text to highlight this point of the referee further.

Please also see our response to the previous reviewer comment 2 and 3. We agree that there is a greater emphasis on values and behavior than beliefs and norms but this emphasis is no greater than the overall case for VBN theory. Section 3.1, however defines all the terms and even illustrates the role of beliefs and norms in how values influences behavior. Further, we also emphasize the role of beliefs in changing norms and hence water use behavior, when beliefs update as a result of environmental degradation from past water use behavior.

We respect the desire of the referee to use a general concept of the feedbacks between value and behavior and given the paucity of data, VBN theory provides us with a fundamental framework to do that exactly. Section 3.1 explains the VBN theory and defines its components, while Section 3.2 deals with data paucity and to what extent such a theory has been (or can be) implemented in socio-hydrological models. Section 3.1 describes briefly some key differences between values, beliefs, and norms. Please see the lines 240-256 in the revised manuscript.

**5) Reviewer Comment:** You did not give a full explanation of Figure 2, and you did not use main info in Figure 2 in your manuscript either, so I would suggest you delete it.

**Authors Response**: Please see our response to the previous reviewer comment 2-4. The illustration of a Murrumbidgee farmer is in context of Figure 2 (we have now made reference to this in the revised manuscript) while Section 3.2 (Figure 2 now also referenced here) confronts data availability with socio-hydrological models that embed the concepts from VBN theory. So we would like to keep Figure 2, if this is acceptable to the reviewer and editor.

**6) Reviewer Comment:** There is a bit repetition between Section 1, Section 2 and Section 4. Besides our findings in Australia (Wei et al., 2017) which you cited and used the data from, we had published similar findings in China (Xiong et al., 2016). I list it here for your information. Yonglan Xiong, Zhiqiang Zhang, and Yongping Wei. 2016. Evolution of China's water issue framed in Chinese mainstream media. AMBIO 45 (2): 241- 251DOI: 10.1007/s13280-015-0716-y.

**Authors response:** We have minimized the repetition, especially in terms of socio-hydro modeling studies cited. For completeness we also cited the work in China by Xiong et al. (2016). Thank you for bringing this to our notice.

**Reviewer # 2 Xi Chen**
**1) Reviewer Comment:** This paper did a review of socio-hydrology (SH) modeling with a focus on several place based studies. Based on the review, the Authors pointed out the importance of social norms and values in SH models. At the end, the paper proposed potential future pathways of SH models and discussed the challenges to generalize SH models. I have the following comments that I hope the authors could address in the revision.
**Authors response:** We thank Xi Chen for his review. We address all the specific comments below.

**Specific comments:**
**2) Reviewer Comment:** The paper explains the review case studies in multiple sessions with too much details. The focus of the paper should be the knowledge generated from those case studies. Maybe the authors can find a way to generalize the information provided by these studies.
**Authors response:** We follow the reviewer's advice and add a conceptual figure (Figure 6), and a one paragraph synthesis along with it, towards the end of section 4 to generalize the information presented in the section. Please note however that section 4 itself was designed to be a synthesis. Please see lines 393-410 in the revised manuscript.

**3) Reviewer Comment:** In section 2.2, maybe the authors should add the following reference, since this study is also using the idea of community sensitivity to do SH modeling.
Chen, X., D. Wang, F. Tian, and M. Sivapalan (2016), From channelization to restoration: Socio hydrologic modeling with changing community preferences in the Kissimmee River Basin, Florida, Water Resour. Res., 52, doi:10.1002/2015WR018194.
**Authors response:** This was an oversight. We have now added the reference. Please see the lines 182-186 in the revised manuscript

**4) Reviewer Comment:** Section 2.3: Roobavannan et al. (2017) is still in review, so it is hard to assess the review materials in this manuscript.
**Authors response:** Roobavannan et al. (2017) is now published and is accessible via http://www.doi.org/10.1002/2017WR020671. We have also updated the citation in the reference list in the revised manuscript.

**5) Reviewer Comment:** Line 288-294: The paper suggested that environment awareness and community sensitivity are both following the general logic of the VBN theory. So maybe the authors can unify the norm/value parameters to one and provide a clear definition based on the VBN theory.
**Authors response:** The purpose of community sensitivity and environmental awareness variable is to capture the society's changing value and norms and follow principles of VBN theory. It should be noted these variables include the values, beliefs and norms together. We have revised the paper to unify the relevant terms and use variables. We agree that we need to further differentiate the variables as we begin to reliably observe them. Please see lines 393-430 in the revised manuscript.

**6)Reviewer Comment:** Line 448-453: van Emmerik et al. (2014) uses environment awareness, not community sensitivity.
**Authors response:** This has been corrected in the revised manuscript.

**7) Reviewer Comment:** Line 511-513: For the three listed river basins, please add the countries they are located in.
**Authors response:** The country of respective river basins are added in the revised manuscript.

**8) Reviewer Comment:** Line 521: Typo: "Elshafiei". These three references have been repetitively mentioned in this manuscript over 10 times. I think the focus of the paper should be the scientific knowledge that can push SH modeling forward, not the three case studies.
**Authors response:** The typo has been corrected. We agree with the reviewer that this section may give the impression that we are just repeating three different case studies. Our intention was really to connect them to VBN theory. We have minimized the apparent repetitions in the revisions.
Indeed, the focus of this section was to highlight the need to include changing values and norms of society in order to predict future projections. Through this review we explain that recent SH model studies have moved closer toward integration with key social science theories of perception and behavior, and have taken steps toward endogenizing values and norms. We intended to show that these models are internally consistent with patterns observed with proxy data of environmental awareness and water policy change, such as the newpaper article-based proxies of Wei et al. (2017). However, such proxy-reliant models are only the beginning of the way towards generalized models and their use in predictions for sustainable water management.

**9) Reviewer Comment:** Figure 5: The paper spends a fair amount of paragraphs to talk about the parameter "community sensitivity", but the analysis provided by the study is using "environment awareness", which I believe is a different parameter. Following my previous comment, maybe the authors should add explanations about the differences between these two parameters and try to generalize the parameters, which would be a part of the SH generalization process.
**Authors response:** Community sensitivity and environmental awareness are variables defined to capture the changing values and norms in different socio-hydrological models. Community sensitivity is an advance over the previously defined environmental awareness. We agree that they are different in the way they are defined, yet both are intended to capture the same concept of changing values and norms for use in socio-hydrological models. We however have added text on the difference between their definitions as the referee suggests, that community sensitivity is a more complex description of environment awareness. Both are modeled as memory variables. But while in the case of latter the time scale of the memory of past environmental disaster is kept constant, in the case of the former (i.e. community sensitivity) the time scale is dynamic and depends on community norms in context of its water environment. Please see lines 201-209.

**NORMS AND VALUES IN SOCIO-HYDROLOGICAL MODELS**

Mahendran Roobavannan[1], Tim H. M. van Emmerik [2], Yasmina Elshafei[3], Jaya Kandasamy[1], Matthew R. Sanderson[4], Saravanamuthu Vigneswaran[1], Saket Pande[2], and Murugesu Sivapalan[5, 6]

[revised manuscript text omitted]

Several place-based socio-hydrology studies in basins dominated by agricultural development, such as the Tarim (China, Liu et al., 2014), Murrumbidgee (Australia, Elshafei et al., 2014; van Emmerik et al., 2014), and Lake Toolbin (Australia, Elshafei et al., 2015) basins, have highlighted a shift in water use behavior from an initial focus on agricultural production to an increasing emphasis on environmental conservation, a shift that has been called the pendulum swing (Kandasamy et al., 2014). Similarly Chen et al., (2016) showed a shift in water management from flood mitgation to wetland protection at Kissimee river, USA. Socio-hydrology models developed to reproduce these observed dynamics attributed the shift to changing human values and norms, which were tracked indirectly through proxies (e.g., environmental degradation). For example, van Emmerik et al. (2014) modeled the human decision to allocate more or less water to agriculture or to the environment on the strength of a dynamic 'social' state variable called environmental awareness, which reflected societal perceptions of the environmental degradation within the prevailing value systems or culture (see also Di Baldassarre et al. (2013) for awareness of floods in the context of coupled human-flood systems, and Garcia et al., (2016) for awareness of shortage for town water supply in the context of coupled human-town water supply 
[revised manuscript text omitted]

Similarly, the community sensitivity concept was used to explain the shift in values and norms, and  management emphasis from flood mitigation to environment protection in Kissimee river, Florida, USA (Chen et.al, 2016). They used wetland storage and flood intensity as proxy to measure changing value system. Their study showed that the value system was affected by the relative size of population in upstream and downstream portions of the catchment.

**Economic diversification and institutions:**

Roobavannan et al. (2017) presented a rigorous validation of the community sensitivity concept of Elshafei et al. (2014) and further extended it to account for the relative dependence of the basin economy on agriculture. Roobavannan et al. (2017) assumed that the tradeoff between economic wellbeing and environmental health at the community level depends also on contextual factors such as economic diversification. In this way the resulting SH model was able to explain the importance of economic diversification and sectoral transformation on the community sensitivity that then impacted human water management actions.

Roobavannan et al. (2017) also introduced a fish spices richness (FSR) index (Yoshika*wa et al*, 2014) as a separate proxy for ecosystem health. They also used time series of economic development (measured by total irrigated area and irrigation water utilisation) and diverse proxies for technology (i.e., patents) and water use behavior (e.g., environmental behavior based on fish species richness index) in validating the dynamic changes to community sensitivity.

Community sensitivity and environmental awareness are different in the way they are defined, yet both intend to capture the same concept of changing values and norms for use in socio-hydrological models. Environmental awareness accounts for society's perception of environment degradation while community sensitivity accounts for the balance of perception of environment degradation and economy growth of a region. Community sensitivity is a more complex assessment variable than environment awareness. Both are modeled as memory variables. But while in the case of latter the time scale of the memory of past environmental disaster is kept constant, in the case of the former (i.e. community sensitivity) the time scale is dynamic and depends on community norms in context of its water environment.

**3. VALUES, BELIEFS AND NORMS AS DYNAMIC VARIABLES**

So far in SH modelling research, aspects of human culture that drive human behaviour in respect of water management – i.e., values and norms – have been treated in a lumped way, represented by proxies, in a black-box manner. Moving SH forward requires opening the 'black box' of culture by questioning the assumptions behind and more clearly measuring and modelling cultural factors. For example, if values are conceptualized as over-arching goals of society (Wescoat, 2013), are they individual goals or collective goals associated with the emergent structure of a coupled human-water system, or both? Similarly, how malleable are values and norms as aspects of a coupled human-
water system? Moreover, under what conditions should values and norms be expected to change, or
remain stable? For that matter, what are the mechanisms through which values and norms might
change, and the human behaviours and actions that result from them?

The ingredients for understanding the role of changing values and norms in coupled human-
water systems can be summarized as (a) forward loop: theories of how individual values influence
individual norms and behavior regarding water use, (b) backward loop: theories of why and how
collective behavior can engender change in individual norms regarding the use of water for
agriculture or the environment, (c) role of institutions in enabling changes in water policy that reflect
collective behavior towards the water environment, (d) data that can provide information on proxy
variables including environment related behavior and patterns and (e) models that use proxy data to
conceptualize processes (a)-(c) in interpreting related patterns. Future work in SH will necessarily
grapple with these types of questions that further elucidate the role of values and norms in coupled
human-water systems.

**3.1 Values, Beliefs, and Norms: VBN theory**
One line of conceptualization seems particularly promising for moving forward socio-hydrological
research. The Values-Beliefs-Norms (VBN) theoretical framework (Stern et al., 1999; Ives and
Kendal, 2014) is grounded firmly in social-psychological theory and has been empirically tested as
a framework for understanding how cultural factors (i.e., values, beliefs and norms) shape
environmental decision-making, and water use behaviour in particular, in a wide array of contexts.
Figure 2 presents a stylized version of a VBN model linking values, beliefs, norms, and behaviours.
In the social sciences, "values" can have various meanings and definitions (Dietz, 2015). The
social science literature on values is voluminous, but there is a large strand of research that employs
the meaning of values from Schwartz (2001: 521), which defined values as "as desirable, trans-
situational goals, varying in importance that serve as guiding principles in people's lives". Values
in this sense are different from beliefs and norms. Beliefs are ideas about what is true (or not); beliefs
can be held regardless of empirical evidence. Norms are rules, written/formal or unwritten/informal
that prescribe behaviors. Norms specify how people should or should not act. Values – as guiding
principles – motivate beliefs and norms, and influence whether people accept particular beliefs and

[revised manuscript text omitted]

Community sensitivity and environmental awareness are defined to capture the changing values and norms in different socio-hydrological models. In van Emmerik et al. (2014) a simple memory function governed by wetland storage sufficed, whereas in Elshafei et al. (2014) more complex community sensitivity equations were introduced, both linking water use related values, beliefs, norms and behavior through two-way feedbacks. Roobavannan et al. (2017), advanced this a step further by representing community level belief about the environment, i.e., community sensitivity, as a consequence of the distribution of weights that individuals attach to enviro-centric versus anthropo-centric values. Such a distribution was made contextual, by making it dependent on economic diversification. The endogenous treatment of values and norms by these studies have implicitly followed the general logic of elements of the VBN theory presented above, even if this was originally unintended (see the feedback from actions to beliefs in Figure 2), and have therefore responded to the challenges of incorporating feedbacks from water use behavior to beliefs and water management norms, consistent with the notion of endogenous and dynamic culture (Caldas et al. 2015).

It should be noted that these variables include values, beliefs and norms together (Figure 6). Indeed there is a need to further distinguish and differentiate the variables as they become more reliably observed, which would be realized with progress in SH. A more generalized understanding of community sensitivity can then be developed.

The pathway to generalisation of SH models is an important goal that allows future prediction (extrapolation in time) and translation of SH models at other geographical locations (extrapolation in space). It provides an important means for the adoption of socio-hydrology in the practice of long-term or strategic water resource management. Generalisation needs to address both the proxies used in SH modelling and the data used to calibrate them, as recent SH modelling studies have highlighted.

Models provide languages or templates in terms of which the following three aspects can be interpreted: 1) how beliefs and norms depend on values, 2) how values and norms influence individual behavior towards the environment, e.g., the wetland health or releasing environment water for bio-diversity, and 3) how pro-environmental behaviour of some in the community (e.g., rallies by the Green Movement) can influence the beliefs of others in the basin and bring about a change in water management (i.e., the feedback). Such templates also enlighten us with variables that need to be measured, so that multiple concepts via the models can be tested and can improve our system understanding.

For example, the policy change in the 1990s in MRB led to increased environmental flow. To interpret this in terms of change in water management norms of the MRB, models need to link beliefs and norms to water use behaviour within the basin. This needs information on a range of relevant values such as altruistic values (i.e., healthy MRB for present and future generations, enough money for the next generation) and egoistic values (i.e., making money), along with information on beliefs, norms, and behaviours, such as how water is being used.

**4.1 Measurement of changing norms and values**

Direct measurement of social value is often very difficult, resulting in the use of indirect methods
(or proxies). Studies have attempted to understand social values on pro-environmentalism
(Bengston 1994; Ives and Kendal 2013) and could be differentiated based on the method of
measurement. Assigned values can be expressed in either monetary or non-monetary terms, and
are relevant to economic and psychology approaches. In a social science context, assigned values
have been quantitatively measured using a variety of techniques, including survey and interview
approaches with the help of psychometric scales used in psychology (Bengston, 1994), social
experiments in behavioral economics (Janssen et al., 2014; Yu et al., 2016) and content analysis
(Seymour et al., 2010; Bark et al. 2016a; Xu and Bengston 1997; Wei et al., 2017).

Schwartz's framework (Schwartz, 1992) specifies a set of ten distinct values across
cultures, which suggests that these are universal motivations for attitudes and behaviors. Humans
differ mainly in terms of the importance attached to the constellation or structure of these held
values. Drawing on Schwartz's framework, values can be measured through survey instruments,
which include items that assess the degree to which respondents feel each value is important for
their life (Dietz et al., 2005). Following Stern et al. (1999), beliefs (general and specific) and norms
– along with Schwartz's value types – have been measured using survey instruments in a wide
array on spatio-temporal contexts. These survey-based measures provide cross-sectional
indicators. Whether, and how, values, beliefs and norms are dynamic is an active area of incipient
research.

Economic valuation offers another set of useful approaches to inform natural resource
management (Farber et al., 2002; Pande et al., 2011; Loomis et al., 2000; Norton and Noonan, 2007;
Wilson et al., 1999; Bark et al., 2016b). Economic valuation approaches to measuring values are
quite distinct from the broader meanings and uses of 'values' described above. These approaches
include non-market valuation (Smith, 1993), contingent valuation (Bateman et al., 2006 ) and other
related techniques, which have been extensively used over the decades and enabled the exploration
of how people 'trade-off' their values in decision-making (Freeman 1993). This enables (i) values
to be measured for large and diverse groups of people, (ii) changes in values to be tracked across
groups of people or across time, and (iii) models to be developed to predict values based on other
factors (e.g., demographics, cultural background). One key limitation of these approaches is that to
the extent that values are measured monetarily, these approaches may not accurately capture
underlying values that are difficult to assign monetary value, but may be more fundamental for decision-making. More generally, there are still unresolved and important questions about value measurement that reflect conceptual and methodological divisions among social sciences and economics. Overcoming these divisions will be crucial for addressing problems in coupled human- water systems.

[revised manuscript text omitted]

112, 8157–9.

Chen, Xi, Dingbao Wang, Fuqiang Tian, and Murugesu Sivapalan. 2016. From Channelization to
Restoration: Sociohydrologic Modeling with Changing Community Preferences in the Kissimmee
River Basin, Florida. *Water Resources Research*, 52,1227-1244. doi:10.1002/2015WR018194.

Di Baldassarre, G., Viglione, A., Carr, G., Kuil, L., Salinas, J.L., Blöschl, G., 2013. Socio-
hydrology: Conceptualising human-flood interactions. Hydrol. Earth Syst. Sci., 17, 3295–3303.

Dietz, Thomas. 2015. Environmental values, in Oxford Handbook of Values, edited by T. Brosch
and D. Sander, pp. 329–349, Oxford Univ. Press, Oxford, U. K.

Dietz, T., Fitzgerald, A., Shwom, R., 2005. Environmental Values. Annu. Rev. Environ. Resour. 30,
335–372.

[revised manuscript text omitted]

**Figure 6.** A conceputal diagram of relationships between variables studied in SH modelling and VBN theory. Black arrows show the feedback loops captured in SH modelling and green arrows show relationships that need to be studied in context of water management.